# Proteomic Analysis of Mucosal and Systemic Responses to SARS-CoV-2 Antigen

**DOI:** 10.3390/vaccines11020334

**Published:** 2023-02-02

**Authors:** Neil Martinson, Bhavna Gordhan, Stefan Petkov, Azure-Dee Pillay, Thabiso Seiphetlo, Natasha Singh, Kennedy Otwombe, Limakatso Lebina, Claudia Fredolini, Francesca Chiodi, Julie Fox, Bavesh Kana, Carolina Herrera

**Affiliations:** 1Perinatal HIV Research Unit, Faculty of Health Sciences, University of the Witwatersrand, Johannesburg 2000, South Africa; 2Department of Science and Technology/National Research Foundation, Centre of Excellence for Biomedical TB Research, School of Pathology, Faculty of Health Sciences, University of the Witwatersrand and the National Health Laboratory Service, Johannesburg 2000, South Africa; 3Department of Microbiology, Tumor and Cell Biology, Karolinska Institutet, SE-171 65 Solna, Sweden; 4School of Public Health, Faculty of Health Sciences, University of the Witwatersrand, Johannesburg 2000, South Africa; 5Affinity Proteomics Stockholm, Science for Life Laboratory, KTH-Royal Institute of Technology, SE-171 65 Solna, Sweden; 6Guys and St. Thomas’ NHS Foundation Trust and King’s College London, London SE1 9RT, UK; 7Department of Infectious Disease, Imperial College London, London W2 1PG, UK

**Keywords:** SARS-CoV-2, vaccine, tissue explants, cytokines

## Abstract

The mucosal environment of the upper respiratory tract is the first barrier of protection against SARS-CoV-2 transmission. However, the mucosal factors involved in viral transmission and potentially modulating the capacity to prevent such transmission have not fully been identified. In this pilot proteomics study, we compared mucosal and systemic compartments in a South African cohort of vaccinated and unvaccinated individuals undergoing maxillofacial surgery with previous history of COVID-19 or not. Inflammatory profiles were analyzed in plasma, nasopharyngeal swabs, and nasal and oral tissue explant cultures, using Olink and Luminex technologies. SARS-CoV-2-specific antibody levels were measured in serum and tissue explants. An increased pro-inflammatory proteomic profile was measured in the nasal compartment compared to plasma. However, IP-10 and MIG levels were higher in secretions than in nasal tissue, and the opposite was observed for TGF-β. Nasal anti-SARS-CoV-2 spike IgG correlated with mucosal MIG expression for all participants. A further positive correlation was found with IP-10 in BioNTech/Pfizer-vaccinated individuals. Systemic levels of anti-SARS-CoV-2 spike IgG elicited by this vaccine correlated with plasma IL-10, IL-6 and HBD4. Proteomic profiles measured in mucosal tissues and secretions using combined technologies could reveal correlates of protection at the mucosal portals of viral entry.

## 1. Introduction

Since the declaration by the WHO on 30 January 2020 of SARS-CoV-2 as a public health emergency of international concern [1], a broad range of clinical outcomes following exposure to the virus have been observed, from asymptomatic to mild non-specific syndromes, to acute respiratory distress syndrome [2]. However, a limited number of studies have evaluated the potential for proteomic biomarkers of risk or protection other than antibody levels in the human mucosal portal of viral entry, the upper respiratory tract [3,4,5,6,7,8,9,10]. Furthermore, despite the rapid approval and roll-out of multiple vaccine strategies [11], which have been key to reducing the number of fatalities, limited knowledge is available on the potential mucosal markers of protection elicited by vaccination, including antibody levels in the nasal and oral cavity, as well as other vaccine-modulated mucosal factors which could affect the susceptibility of these portals of entry to SARS-CoV-2 infection. 

Upper respiratory tract sampling with nasopharyngeal and/or oropharyngeal swabs has been broadly used for COVID-19 testing, with different screening efficacy being reported between both sampling methods [12]. Moreover, protein concentrations in nasal secretions are highly dependent on the collection method [13]. Hence, new tools are required to further characterize the protein content of secretions, and these biological fluids might not fully recapitulate the proteomic profile of mucosal tissues [14,15]. 

Despite the difficulties associated with mucosal tissue sampling, here we highlight how such tissues, cultured as explants, can help define biomarkers of humoral responses, and reveal cytokine/chemokine profiles which are tissue specific within the upper respiratory tract and are different from those observed in secretions. Our study included individuals having tested negative for SARS-CoV-2 prior to surgery, with a history of previous COVID-19 diagnosis or not, and having been vaccinated or not against SARS-CoV-2. Furthermore, we describe how new multiplexing tools such as the Olink platform based on Proximity Extension Assay (PEA) technology, can be used to further characterize mucosal secretions at the proteomic level.

## 2. Materials and Methods

### 2.1. Study Design

The COVAB study aimed to investigate SARS-CoV-2 infectiousness and antibody (Ab) evolution in COVID-19 patients in Sub-Saharan Africa and Europe. A prospective cross-sectional sub-study was set up in South Africa recruiting adults aged between 18 and 70 years old who had a recent SARS-CoV-2 negative test. These individuals consented to providing residual nasal and oral tissue while undergoing elective maxillofacial surgery. Eighteen participants who had a negative SARS-CoV-2 test prior to surgery were recruited, and parameters such as age, HIV status, smoking, previous SARS-CoV-2 infection, and SARS-CoV-2 vaccination (BNT162b2 vaccine by BioNTech/Pfizer, New York, NY, USA, or Janssen/Johnson & Johnson, New Brunswick, NJ, USA, COVID-19 single-dose vaccine by J&J) were recorded (Appendix A). This sub-study was approved by the University of Witwatersrand, Human Research Ethics Committee and the Protocol/Expert Reviewer (Reference 200711). The Swedish Ethics Review Authority approved the laboratory studies of specimens collected in South Africa at the Karolinska Institutet (2021-06076). 

### 2.2. Tissue Explants

Surgically resected oral and nasal tissues were collected in South Africa at Donald Gordon Hospital, Parktown, and Chris Hani Baragwanath Academic Hospital, Soweto. Specimens were immediately transferred into cold Dulbecco’s Modified Eagle’s Medium (DMEM) and transported to the laboratory on ice (median time = 30 min). Upon arrival, specimens were dissected and cut into 2–3 mm^3^ explants comprising epithelial, basement membrane, and lamina propia. Oral and nasal explants were maintained in DMEM/Nutrient Mixture F-12 Ham (DMEM/F-12) containing 2 mM L-glutamine and antibiotics (100 U of penicillin/mL, 100 µg of streptomycin/mL, 0.25 µg of streptomycin/mL, and 80 µg of gentamicin/mL), at 37 °C in an atmosphere containing 5% CO_2_.

### 2.3. Blood Collection and Processing

A single clot activator serum tube (BD, Franklin Lakes, NJ, USA) was obtained, mixed by inversion, and spun down. Serum (twice 1 mL) for Ab levels’ analysis was collected and frozen at −80 °C. Four ACD solution A tubes (BD, Franklin Lakes, NJ, USA) were collected, mixed by inversion, and transferred to LeucoSep tubes (Greiner Bio One, Frickenhausen, Germany) containing Lymphoprep™ (Stem Cell Technologies, Vancouver, BC, Canada). Following density gradient centrifugation, plasma was collected, clarified by centrifugation (1000× *g* for 20 min), and two aliquots of 1 mL each were stored at −80 °C.

### 2.4. Nasal Swabs

One nasopharyngeal swab (Shenzhen Mandelab Co., Shenzhen, China) was collected per participant prior to surgery. The nasopharyngeal (NP) swabs were immediately stored at −80 °C upon arrival at the laboratory. For proteomic analysis, swabs were eluted in 200 µL PBS, spun down 16,873× *g* for 5 min, and protein content was measured via BCA (Bicinchoninic Acid) protein assay (Thermo Fisher Scientific, Waltham, MA, USA). Protein content was in the range of 3–8 mg/mL. 

### 2.5. Multiplex Cytokine Analysis

The levels of 23 cytokines in tissue supernatants after 24 h of culture and in plasma were quantified by in house multiplex bead immunoassay as described previously [16] using a Luminex 100 Systems (Bio-Rad, Hercules, CA, USA). Culture supernatants were spun down to remove any cells or cellular debris prior to processing for Luminex analysis. A minimum of two biological replicates were analyzed.

### 2.6. Proteomic Analysis with the Olink Platform

Plasma samples and nasal swabs were analyzed for the presence of 92 inflammation-related soluble proteins using the Olink Inflammation Panel (Olink Bioscience AB, Uppsala, Sweden) at the Affinity Proteomics Stockholm Unit, Science for Life Laboratory (Stockholm, Sweden) [17]. This platform uses the Proximity Extension Assay technology which is based on a pair of oligonucleotide-labeled antibodies, referred to as probes, which bind to the target protein. A unique PCR target is then formed via a proximity-dependent DNA polymerization event as the probes come in close contact to each other. The new target is finally detected and quantified using qPCR. One technical replicate was analyzed per sample. Protein abundance is expressed as normalized protein expression (NPx) levels.

### 2.7. Anti-SARS-CoV-2 Antibody Detection

Anti-nucleocapsid protein (N) IgG, anti-spike protein (S) IgG and IgM levels were measured in serum with quantitative chemiluminescence microplate immunoassays (CMIA) from Abbott Laboratories (Chicago, IL, USA) using an Abbott Architect i2000SR instrument. Anti-S IgG was also measured in undiluted tissue explant culture supernatants with the same kit. Assays were kindly performed by Dr. Doreen Janse van Rensburg (AMPATH Laboratories, Pretoria, South Africa)

### 2.8. Statistical Analysis

Cytokine concentrations were calculated from sigmoid curve-fits (Prism v. 9.2.0, GraphPad, San Diego, CA, USA). All data presented fulfill the criterion of R^2^ > 0.7. Raw normalized protein expression (NPx) data from the Olink platform were transformed by scaling, yielding values with a mean of 0 and standard deviation of 1. Concentration values as well as NPx data were statistically compared using unpaired *t* test and *p* values. Significance was considered when *p* < 0.05. 

Heat maps were obtained using Prism with cytokine/chemokine levels normalized for protein content measured by BCA to correct for variation in explant size. Fold changes were calculated, and log transformed (base 2). The statistical significance of differences between compared groups was determined using the unpaired, multiple *t* test with no correction for multiple comparison. Differentially abundant proteins were analyzed using Ingenuity Pathway Analysis software (Qiagen, Hilden, Germany) to determine the biological processes affected by vaccination. The pathways with a minimum of at least two analytes associated and *p* < 0.05 were considered to be enriched.

Correlation analysis was performed in R statistical software (v4.1.2; R Core Team 2021) and the correlation matrices of the correlation coefficients between the variables were constructed using Pearson correlation tests with two-tailed *p* values.

## 3. Results

### 3.1. Analysis of the Nasopharyngeal and Systemic Proteomic Profiles

To broaden the proteomic characterization of the upper respiratory tract environment, we evaluated the possibility to use the Olink technology to analyze secretions obtained from nasopharyngeal swabs. The recommended sample protein concentration for Olink analysis ranges between 0.5 and 1 mg/mL. Hence, to assess the sensitivity of the Olink platform with this matrix, NP secretions were tested undiluted and at three serial dilutions (1:4, 1:8, and 1:16). Out of the 92 proteins included in the panel, 14 were below the lower limit of detection for both matrices. Comparison of NPx data from secretions with plasma values confirmed the use of undiluted samples for both matrices, NP secretions and plasma (Appendix A). 

As expected, the protein profile in NP secretions was different from that observed systemically (Figure 1), reaching statistical significance for 79% of analytes included in the Olink panel (Appendix A). Most proteins (76%) were present at higher levels in the NP compartment than in blood (Figure 1a,b). The NPx values for ARTN, IL-1α, IL-20RA, IL-22 RA1, IL-33, and LIF fell below the lower limit of detection for plasma, but could be measured in NP secretions (Figure 1a). Proteins with increased mucosal abundance were linked to the activation of pathways such as pathogen induced cytokine storm, macrophage activation, IL-17 signaling, and wound healing; and to the inhibition of the erythropoietin signaling pathway, LXR/RXR activation and anti-inflammatory IL-10 signaling (Figure 1c). Proteins with significantly reduced mucosal expression included anti-inflammatory proteins (FGF-19 *p* < 0.0001, FGF-21 *p* < 0.0001, IL-10RB *p* < 0.0001), chemokines (I-TAC/CXCL11 *p* = 0.0015, Eotaxin/CCL11 *p* < 0.0001, MCP-2/CCL8 *p* < 0.0001, MPIF-1/CCL23 *p* < 0.0001, TECK/CCL25 *p* < 0.0001, MCP-4/CCL13 *p* < 0.0001), as well as proteins linked to anti-microbial (CSF-1 *p* < 0.0001, CD8A *p* < 0.0001) and inflammatory signaling proteins (IL-12B *p* = 0.0075, SCF *p* < 0.0001, Flt3L *p* = 0.0189, CD6 *p* = 0.0307) (Appendix A).

### 3.2. Nasopharyngeal Secretions Do Not Fully Recapitulate the Proteomic Profile in Nasal Tissue

We then investigated whether the proteomic profile observed in NP secretions was representative of that in nasal tissue. With this aim, the levels of 33 cytokines/chemokines were measured in culture supernatants of nasal tissue explants and compared with those found in plasma (Figure 2). As observed in NP secretions, a greater number of cytokines/chemokines (61 %) were significantly overabundant in nasal tissue compared to the levels found in blood (Figure 2c, Appendix A). However, the concentrations found in plasma were significantly higher for some proteins with anti-microbial (L-selectin *p* < 0.0001, GM-CSF *p* = 0.0035), chemiotactic (MCP-2 *p* < 0.0001, RANTES *p* = 0.0005, SDF-1β *p* = 0.0112) or adaptive activity (IL-4 *p* = 0.0004, IL-15 *p* = 0.0050). Functional pathway analysis revealed enrichment in the nasal tissue of pathways including inflammatory cytokines IL-1α, IL-1β, IL-6, and IL-17 (Appendix A). 

Despite this pro-inflammatory mucosal profile being similar to that observed in NP secretions, we next wanted to determine if the plasma-proportional levels of mucosal cytokine/chemokine found in tissue were comparable to those in secretions. Plasma-normalized levels of expression for most common proteins analyzed in both platforms, Olink and Luminex, were of different magnitude (Appendix A). Interestingly, for some cytokines/chemokines, these levels were opposite. Higher levels of chemokines, IP-10 and MIG, and adaptive cytokine IFN-γ were found in NP secretions than in tissue; however, the proportional level of the anti-microbial cytokine TGF-β was higher in nasal tissue explants than in secretions.

### 3.3. Evaluation of Proteomic Differences within the Upper Respiratory Tract

Oral and nasal tissues were obtained during the same surgery from one unvaccinated patient with undetectable anti-SARS-CoV-2 Abs in serum. As a pilot study, cytokine/chemokine levels were measured in both tissues (Figure 3). Most proteins were found to be expressed at different levels in oral explant culture supernatants compared to nasal tissue (Figure 3a,b); however, only some differences reached statistical significance (Figure 3c, Appendix A). Inflammatory (IL-1β *p* = 0.0175, IL-12 *p* = 0.0359, TNF-α *p* = 0.0241), adaptive cytokines (IL-2 *p* = 0.0260, and IL-15 *p* = 0.0090), and anti-microbial cytokines (GM-CSF *p* = 0.0050, IFN-β *p* = 0.0176) as well as the chemokines RANTES (*p* = 0.0411) and MCP-2 (*p* = 0.0367) were expressed at significantly higher levels in oral tissue than in nasal explants. However, the oral levels of adaptive cytokine IL-17 (*p* = 0.0128) and anti-microbial proteins, HBD4 (*p* = 0.0110) and L-selectin (*p* = 0.0291), were significantly lower than in the nasal compartment. This pro-inflammatory cytokine profile was linked to the activation of biological pathways including pathogen-induced cytokine storm signaling, macrophage activation, the role of hyper-cytokinemia/chemokinemia and of inhibitory MAPK in influenza pathogenesis, the role of pattern recognition receptors of bacteria and viruses, IL-17 and HMGB1 signaling, and wound healing (Appendix A). Pathways linked to erythropoietin and anti-inflammatory IL-10 signaling, VDR/RXR activation, and macrophage alternative activation signaling were inhibited. 

### 3.4. Correlation Analysis of Humoral and Cytokine Responses

To evaluate if the mucosal and systemic cytokine/chemokine profiles could be biomarkers of the B-cell responses elicited in both compartments, we performed a correlation analysis (Appendix A). A limited number of significant Ab-systemic cytokine/chemokine correlations were found (Figure 4). The plasma levels of anti-inflammatory cytokine IL-10 correlated with systemic levels of anti-S IgG in BioNTech/Pfizer-vaccinated participants (Figure 4c); however, a negative correlation was found with nasal anti-S IgG in vaccine-naïve participants (Figure 4b). Furthermore, in these later participants, serum anti-S IgG significantly correlated with systemic levels of IL-6, an inflammatory cytokine, and HBD4, an anti-microbial protein; and serum levels of anti-N IgG correlated with systemic G-CSF expression, an anti-microbial cytokine. 

For all participants, independently of the route of exposure to the viral antigen, levels of anti-S IgG in nasal explant culture supernatants significantly correlated with the secretion of chemokine MIG (Figure 5). In participants immunized with the BioNTech/Pfizer vaccine, further significant correlation of these IgG levels was found with mucosal levels of chemokine IP-10 (Figure 5c). In vaccine-naïve participants, in addition to MIG, positive correlations were found between nasal anti-S IgG and nasal anti-microbial proteins (GM-CSF, L-selectin, and elafin), inflammatory cytokines (IL-1β and IL-12), chemokines (MIP-1β and SDF-1β), and adaptive cytokine IL-16. 

Serum anti-S IgG levels correlated with nasal RANTES in vaccine-naïve participants; however, in BioNTech/Pfizer-vaccinated individuals, serum anti-S IgG correlated with nasal TGF-β and was inversely proportional to nasal anti-microbial proteins G-CSF, HBD4, and *p*-selectin. Serum anti-N IgG also inversely correlated with nasal expression of anti-microbial cytokines G-CSF and HBD4 in vaccinated individuals. In these participants, further negative correlations were found with IL-7, while IL-2 directly correlated with serum anti-N IgG levels.

Serum IgM inversely correlated with nasal G-CSF, MCP-2, IL-4, and IFN-β in vaccine-naïve individuals. Interestingly, levels of serum IgM elicited following BioNTech/Pfizer-vaccination only one correlation, inverse, which was found with nasal IFN-γ.

Despite the limited number of participants included in this study, among the BioNTech/Pfizer vaccinated and non-vaccinated groups there were individuals with different smoking status (current smokers and non-smokers). Inclusion of this parameter in the correlation analysis revealed significant positive correlations between smoking and serum levels of anti-N IgG in vaccinated participants (Figure 4c or Figure 5c). In this group of individuals, smoking also correlated significantly with expression of IL-2 in nasal tissue explants (Figure 5c) and anti-microbial proteins SLP-1 and HNP-1-3 in plasma (Figure 4c). However, a negative correlation was found with plasma G-CSF. In vaccine-naïve participants, the positive correlation between smoking and plasma HNP-1-3 was also observed (Figure 4c). A further significant correlation was found with plasma levels of IL-16. In this group, a non-significant inverse correlation was revealed between smoking status and nasal tissue anti-S IgG levels (Appendix A).

## 4. Discussion

Understanding the proteomic and humoral responses that are elicited at the mucosal portals of viral entry, following infection-induced or vaccine-mediated immunization against SARS-CoV-2 could provide insight into factors that determine susceptibility to or protection against infection, and potentially, clinical outcome following viral exposure.

Several studies have assessed the mucosal immune responses elicited within the upper respiratory tract in SARS-CoV-2-infected individuals with distinct clinical classification and compared them with those induced systemically. Smith et al. reported a lack of correlation between both compartments for Abs and cytokines/chemokines [8]. The plasma viral load was directly associated with systemic pro-inflammatory cytokines and spike-specific neutralizing Abs; however, nasopharyngeal viral load inversely correlated with mucosal IFN responses. Comparing nasal washes with serum in asymptomatic and symptomatic SARS-CoV-2-infected individuals, Ravichandran et al. also found compartment-specific Ab responses. However, they reported higher levels of IFN-α in nasal washes from symptomatic COVID-19 patients [18]. Compartmentalization of Ab responses has also been described in convalescent individuals [19,20] and following immunization with the BioNTech/Pfizer vaccine [21,22].

Here, we report for the first time the use of the Olink platform for broader proteomic analysis of NP secretions obtained with nasopharyngeal swabs. Eighty-five percent of analytes included in the platform were detected in undiluted secretions. The titration study showed NPx values proportional to the sequential dilutions tested and for most analytes, the values obtained with the highest dilution (1:16) were still above the lower limit of detection. However, different plasma-normalized proteomic profiles were observed in NP secretions with Olink, and in nasal explant culture supernatants with Luminex; with opposite proportions for proteins such as IP-10, MIG, IFN-γ, and TGF-β (Appendix A). Hence, the analysis of NP secretions does not fully recapitulate the immune responses elicited in mucosal compartments as previously suggested [14,15]. Furthermore, our preliminary comparative analysis of the oral and nasal environment in one patient (Figure 3) encourages further investigation in larger cohorts to define tissue-specific factors involved in viral transmission and in vaccine-induced protection. These could be used to design new preventive/therapeutic strategies or to inform the design of a new vaccine.

Several methods can be applied for the proteomic analysis of nasopharyngeal swabs and a large number of studies in the field have been published applying mass spectrometry, reviewed in [23,24,25,26]. The present study highlights the possibility of measuring the proteomic profile of nasopharyngeal swabs using a technique, Olink, which is suitable for measuring proteomics in a large number of samples, within a short time and at limited costs. A comparison between mass spectrometry and Olink is not available because measurement of over 90 protein targets in mass spectrometry is not a realistic goal; in addition, the size of collected material would not allow for this comparison.

The proteomic profile in tissue explants could be affected by the specimen processing including surgical resection and the cutting of explants, which trigger inflammatory responses. To minimize the impact of the procedure, we measured cytokine/chemokine levels after 24 h of culture, a time point that allows the detection of secreted proteins into the culture media when the background of inflammation is still low [27]. 

We found limited cytokine–Ab correlations in systemic samples from BioNTech/Pfizer-vaccinated individuals, which has also been reported by Peng et al. [28]. However, the distinct cytokine–Ab correlations observed systemically (Figure 4) could potentially differentiate immune responses induced upon infection from those elicited by vaccination. In the upper respiratory tract, nasal anti-S IgG levels were directly correlated with MIG independently of the route of exposure to the viral antigen, infection, or vaccination. However, other distinct nasal cytokine-nasal anti-S IgG correlations revealed potential patterns specifically-associated with prime BioNTech/Pfizer vaccination (positive correlation with IL-10) in contrast to those found in vaccine-naïve participants (positive correlation with GM-CSF, L-selectin, elafin, IL-1β, IL-12, MIP-1β, SDF-1β, and IL-16). Mucosal pro-inflammatory cytokines and anti-microbial proteins have been linked with an efficacious early mucosal immune response against SARS-CoV-2 [4].

Among the populations considered to have a higher risk of severe clinical outcome following exposure to SARS-CoV-2, smoking has been highlighted as a contributing factor [29]. This is thought to be linked with the upregulated baseline expression of IL-6, TNF-α, and pro-inflammatory cytokines in lung due to chronic smoking [30]. Vaccine-induced responses have also been reported to be affected by smoking, with lower Ab titers elicited and faster IgG waning [31]. Our correlation analysis revealed a significant direct link between smoking and serum anti-N IgG levels in BioNTech/Pfizer-vaccinated participants, and a non-significant correlation with nasal anti-S IgG levels in the vaccine-naïve group. However, this result could be affected by the low number of self-reported current smokers. Other aspects, such as HIV status, age, diabetes, BMI, hypertension, or asthma, have been evaluated as potential risk factors of severe clinical outcome [4]. We collected this information for all participants enrolled in this sub-study (Appendix A). None of the participants were living with HIV; however, this pilot study supports further analysis evaluating the potential impact of the baseline characteristics on the mucosal environment profile. 

This preliminary study has several limitations. First, a low number of participants were enrolled in the study and recruitment occurred at an advanced phase of the pandemic in South Africa, after the roll-out of vaccination. As a consequence, only one participant could be considered SARS-CoV-2 naïve, not previously infected nor vaccinated, based on serum Ab levels. Among the vaccinated individuals, only one participant received the single dose of the J&J vaccine; therefore, no correlation analysis was possible. However, 11 participants had received a first BioNTech/Pfizer vaccine dose 19.8 ± 14.9 weeks before surgery. Second, the comparison between oral and nasal compartments could only be explored with specimens from one individual who had no detectable SARS-CoV-2 Abs in serum, and therefore no further conclusions other than observing the expected differences between both mucosal compartments within the upper respiratory tract could be reached. Third, swabs and tissue explant culture supernatants could not be analyzed by both Olink PEA and Luminex multiplexing technologies due to sparse sample availability. Hence, no direct comparison of proteomic profiles between secretions and nasal explants could be established, but normalization towards plasma data was used. Furthermore, no oral swabs were collected. In future studies it will be important to measure IgA levels in both compartments, mucosal and systemic. Finally, the kits used to measure anti-S IgG levels in explant culture supernatant had not been optimized for this matrix due to lack of available samples for such optimization.

This research communication intends to highlight the potential of the tissue explant models combined with multiplex assays for proteomic analysis; this approach could be useful to study the immunopathology of emerging viruses at the mucosal portals of viral entry.

## 5. Conclusions

The distinct inflammatory profile and local responses to antigen exposure (summarized in Figure 6) highlight the importance of collecting mucosal samples to understand the local responses against mucosally-transmitted pathogens and to evaluate the efficacy of preventive and therapeutic candidates. The mucosal proteomic profiles described in this pilot study support further investigations to refine the proteomic patterns associated with risk and protection, and to define biomarkers of previous SARS-CoV-2 infection.

## Figures and Tables

**Figure 1 vaccines-11-00334-f001:**
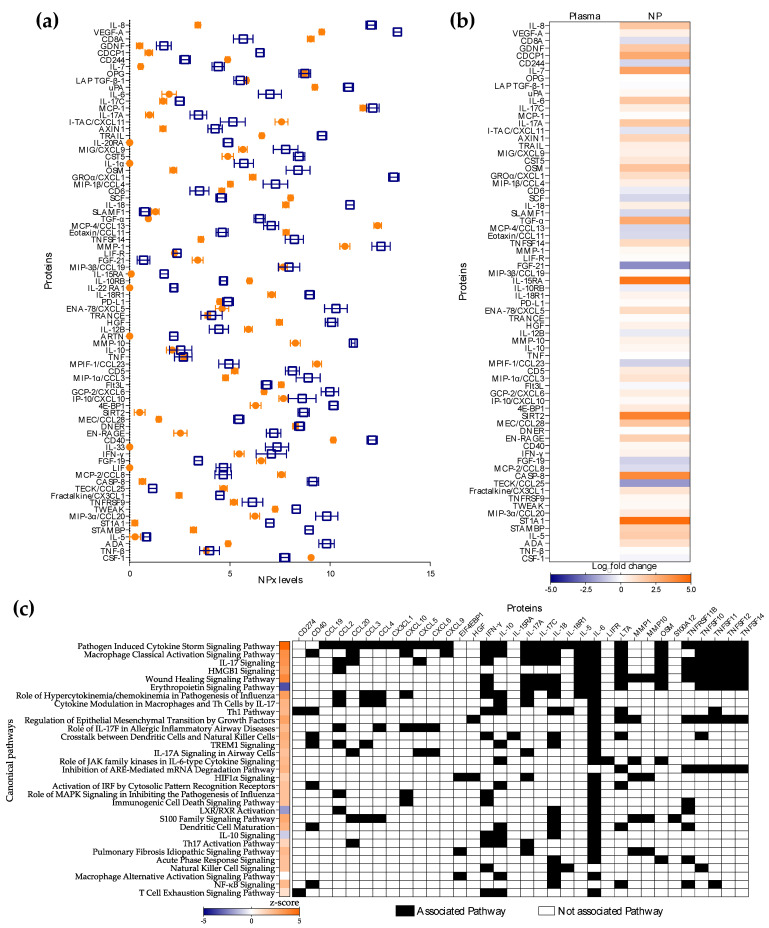
Comparison of systemic and nasopharyngeal (NP) secretion proteomic profiles. Plasma and nasal swabs were obtained from each study participant. Plasma was analyzed undiluted. NP secretions were collected from swabs and analyzed undiluted. (**a**) Normalized protein expression (NPx) levels in plasma (
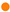
) and NP secretions (
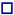
) were measured via Olink technology. Comparative Luminex/Olink nomenclature is shown. (**b**) Heatmap representing upregulated (orange) or downregulated (blue) proteins in NP secretions vs. plasma. Differences are shown in Log_2_ from single replicates for each participant. (**c**) Canonical pathways significantly enriched in NP secretions in comparison to plasma. The activation z-scores determined by directionality (overabundance in orange and underabundance in blue) and number of proteins are represented by a heat map. The proteins associated to each pathway are shown in black cells. The pathways included had a minimum of two associated analytes and a *p* < 0.05 (right-tailed Fisher exact test).

**Figure 2 vaccines-11-00334-f002:**
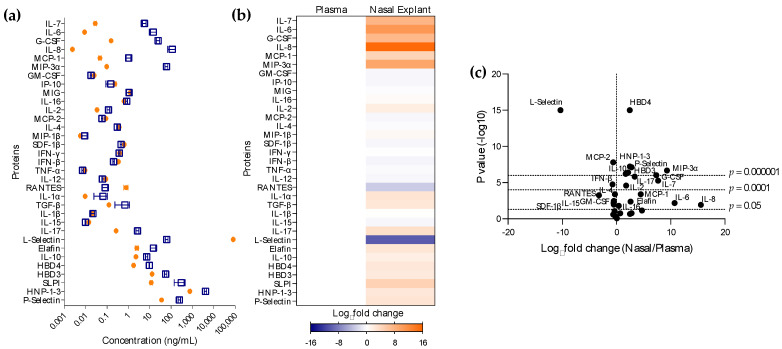
Comparison of systemic and nasal tissue proteomic profiles. Plasma and nasal tissue were obtained from each study participant. Tissues were cut into explants and cultured for 24 h. (**a**) Cytokine expression levels in plasma (
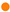
) and nasal explant culture supernatants (
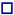
) measured via Luminex technology. (**b**) Heatmap representing upregulated (orange) or downregulated (blue) cytokines in nasal explants supernatants vs. plasma. Differences are shown in Log_2_ from triplicates for each participant. (**c**) Volcano plots comparing cytokine secretion in culture supernatants of nasal explants to plasma. Analytes above horizontal dotted line corresponding to a *p* value of 0.05 are significantly modulated. Vertical dotted line is set at a Log_2_ fold change = 0.

**Figure 3 vaccines-11-00334-f003:**
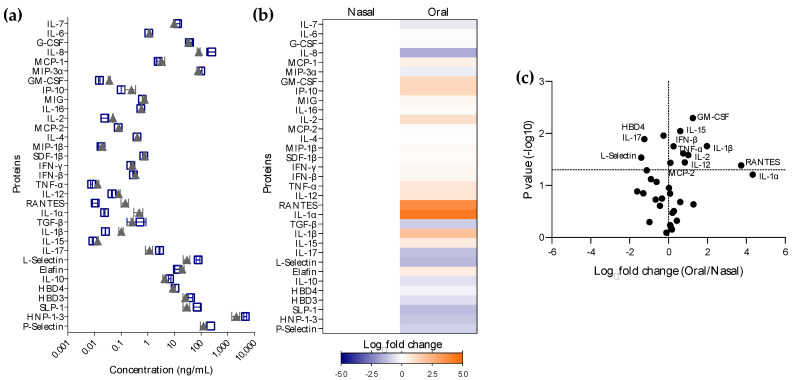
Mucosal cytokine profiles in the upper respiratory tract. Nasal and oral tissues were obtained from a non-vaccinated participant with no prior exposure to SARS-CoV-2 based on Ab levels. Tissue explants were cultured for 24 h. (**a**) Cytokine expression levels in nasal (
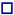
) and oral (
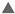
) culture supernatants were measured via Luminex technology. (**b**) Heatmap representing upregulated (orange) or downregulated (blue) cytokines in culture supernatants from oral vs. nasal explants. Differences are shown in Log_2_ from one experiment performed in triplicates. (**c**) Volcano plots comparing cytokine secretion in culture supernatants of oral explants to nasal tissue. Horizontal dotted line indicates a *p* value of 0.05. Analytes above this line are significantly modulated. Vertical dotted line is set at a Log_2_ fold change = 0.

**Figure 4 vaccines-11-00334-f004:**
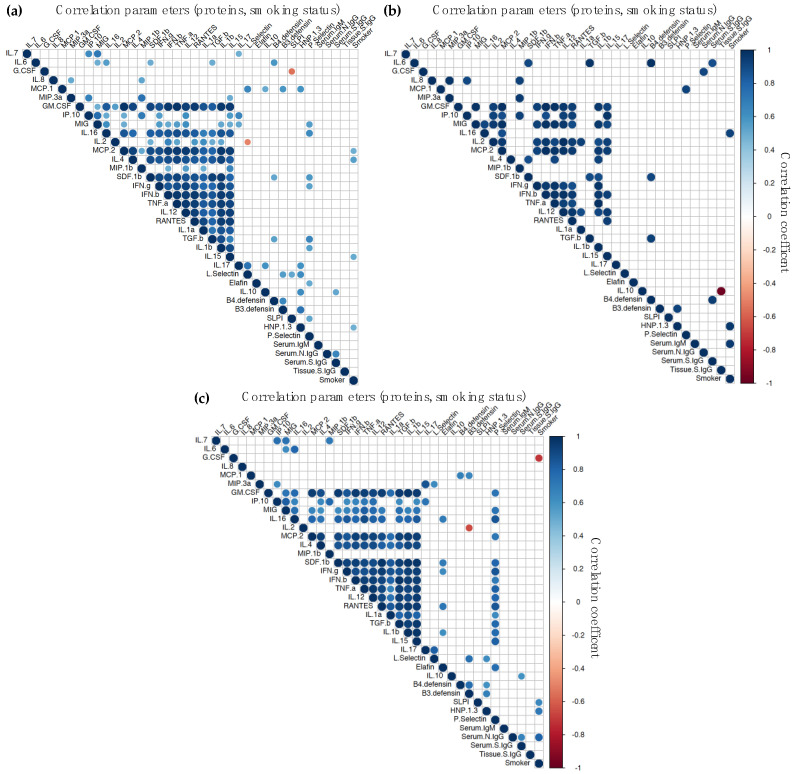
Correlogram of significant responses elicited systemically. Significant (*p* < 0.05) correlations between concentrations of plasma cytokines/chemokines, serum and nasal tissue Ab levels, and smoking status for all participants (**a**), vaccine-naïve (n = 6) (**b**) or BioNTech/Pfizer-vaccinated participants (n = 11) (**c**) were visualized by a correlation matrix. Positive Pearson correlation coefficients are displayed in blue and negative correlations in red color. The color intensity indicated in the side bar is proportional to the correlation coefficient. The size of the circle is proportional to the statistical significance of the correlation.

**Figure 5 vaccines-11-00334-f005:**
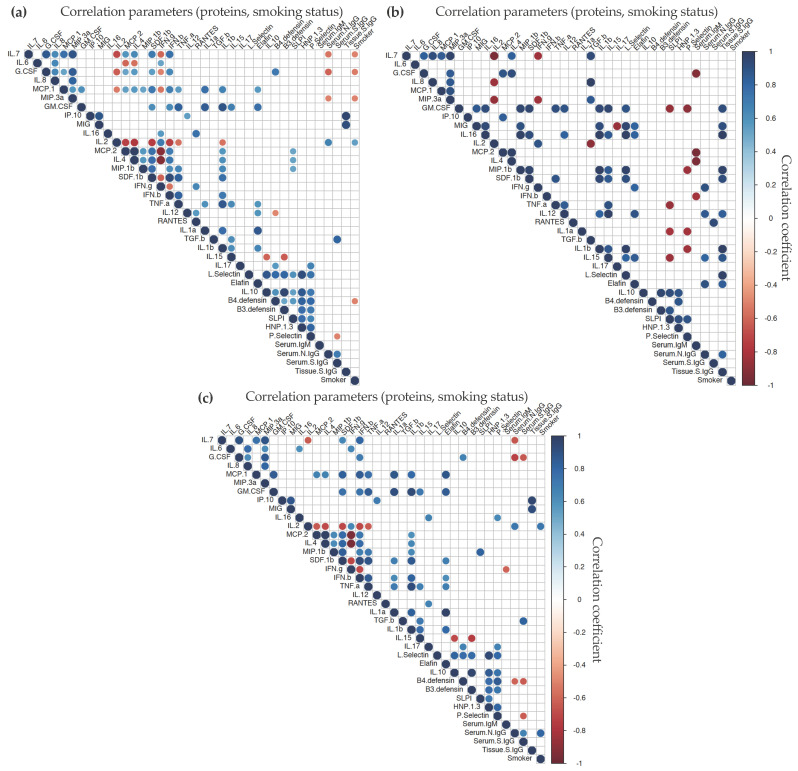
Correlogram of significant responses elicited mucosally. Significant (*p* < 0.05) correlations between concentrations of cytokines/chemokines in nasal explant culture supernatants, serum, and nasal tissue Ab levels and smoking status for all participants (**a**), vaccine-naïve (n = 6) (**b**) or BioNTech/Pfizer-vaccinated participants (n = 11) (**c**) were visualized by a correlation matrix. Positive Pearson correlation coefficients are displayed in blue and negative correlations in red color. The color intensity indicated in the side bar is proportional to the correlation coefficient. The size of the circle is proportional to the statistical significance of the correlation.

**Figure 6 vaccines-11-00334-f006:**
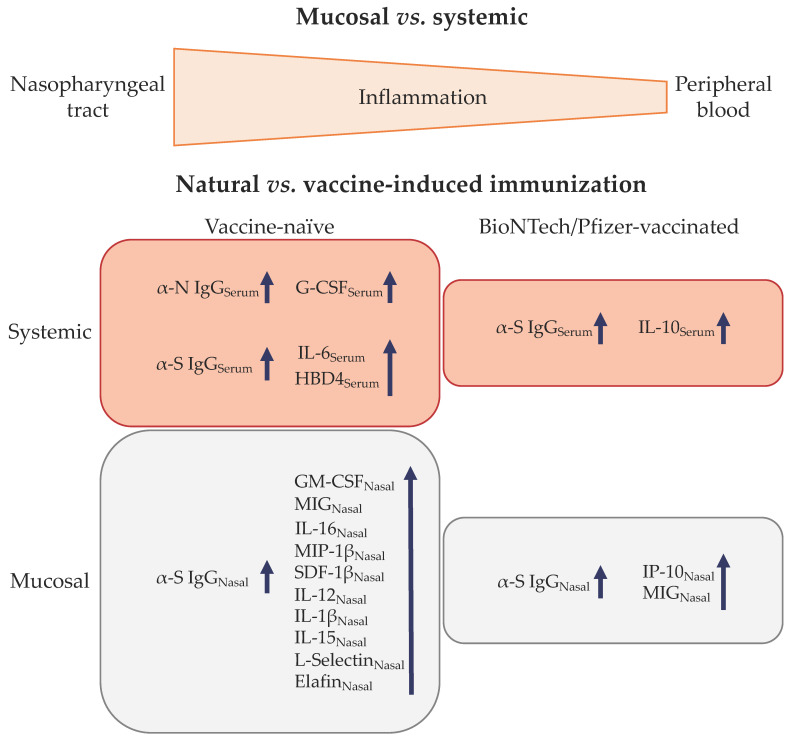
Comparison of nasopharyngeal and systemic environment. Greater inflammatory background was observed mucosally rather than systemically; and correlations between local responses (antibodies vs. cytokines/chemokines) are compartment specific and antigen dependent. Blue arrow indicates sense of correlation between antibodies and cytokines/chemokines; i.e., both arrows in the same direction indicate a positive correlation.

## Data Availability

Data are available upon request.

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
