# Peer review of "Proteomic Analysis of Mucosal and Systemic Responses to SARS-CoV-2 Antigen"

_vaccines, 2023, doi:10.3390/vaccines11020334_

Round 1

Reviewer 1 Report

There are a number of questions and comments that the authors should carefully address:

1. It would make sense to explain in distinct correlations how the insights gained from the expression profiles of many genes under different conditions relate or could relate to the immunological defense against SARS-CoV-2.

2.  To what extent might the explant of tissues affect activity profiles of indivdual genes?

3. Inclusion of the breakdown of data from smokers versus non-smokers is confusing and dilutes the flow of arguments. What was the idea behind this aspect of the study?

4. Obviously, tissues from patients and controls could have been obtained only by surgery of whatever stage of difficulty. It remains unexplained what the reasons have been for removal of tissues from Covid-19 patients or from control indviduals. It can be suspected that the reason for surgery could have been ailments unrelated to virus infection. If so, this would render the interpretation of data extremely difficult. There would then be too many parameters to render the analyses controllable. Moreover, surgical interference could affect activity profiles.

5. The legend to each figure should present information on the number of probands studied in each experimental analysis.

6. I recommend summarizing the data on mucosal versus serum gene activities under different conditions and support this comparison by a graph or graphs that will help the reader to grasp the essential message of this article.

7. Similarly, a summary that clearly juxtaposes serum versus tissue activities will have to be added to the article. You cannot expect the reader to decipher each of the highly complex graphs for certain activities and arrive at  meaningful conclusions.

Minor points:

Line 39 - Sentence should read: ..... could reveal correlates of protection ....

Figures 1c, 4a to c and 5 a to c: Please improve the graphic designation of genes. It is very difficult to decipher individual entries of nomenclature.

Author Response

Response to Reviewer 1 Comments

There are a number of questions and comments that the authors should carefully address:

  1. It would make sense to explain in distinct correlations how the insights gained from the expression profiles of many genes under different conditions relate or could relate to the immunological defense against SARS-CoV-2.

This Olink platform is not a transcriptomic or genomic methodology, but a proteomic technique. The modulation of pro-inflammatory cytokines/chemokines and anti-microbial proteins has been shown to be linked with efficacious early mucosal immune response against SARS-CoV-2. We have added a sentence and reference in the discussion (lines 363-365).

  1. To what extent might the explant of tissues affect activity profiles of individual genes?

We agree with the reviewer that processing of tissue explant, including surgical resection and cutting of explants, induces an inflammatory response that is observed at the proteomic and transcriptomic levels as we have previously described in other tissues using the same methodology (Herrera et al., Vaccines 2021). However, in the current study no transcriptomic nor genomic analysis was performed. The harvesting point chosen for the proteomic analysis is routinely used in our and other laboratories to be able to measure proteomic responses without significant masking by the increasing inflammatory background as explained in the discussion (lines 349-353)

  1. Inclusion of the breakdown of data from smokers versus non-smokers is confusing and dilutes the flow of arguments. What was the idea behind this aspect of the study?

Taking into account that this is a communication, we wanted to do a pilot evaluation of the potential link between participant characteristics and proteomic responses to SARS-CoV-2 antigen. We chose the smoking status as a starting point for assessment considering the existing literature describing the effect of smoking on the inflammatory baseline of the respiratory tract and on the vaccine efficacy. We have added a section in the discussion highlighting that future correlation analysis will be performed with the additional participant characteristics collected during enrolment (lines 375-380).

  1. Obviously, tissues from patients and controls could have been obtained only by surgery of whatever stage of difficulty. It remains unexplained what the reasons have been for removal of tissues from Covid-19 patients or from control individuals. It can be suspected that the reason for surgery could have been ailments unrelated to virus infection. If so, this would render the interpretation of data extremely difficult. There would then be too many parameters to render the analyses controllable. Moreover, surgical interference could affect activity profiles.

No COVID-19 patients were included in the study, participants had to test negative prior to surgery as stated in line 77; however, history of SARS-CoV-2 infection and vaccination were collected from each participant. As stated in the section “study design”, the individuals included in the study consented to provide residual nasal and oral tissue while undergoing elective maxillofacial surgery This has been further clarified in the introduction (lines 66-68) and in the methods sections (line 79).

  1. The legend to each figure should present information on the number of probands studied in each experimental analysis.

We agree that specifying the number of participants in each analysis helps the reader and have updated the legends of Figure 4 and 5 to include the specific number of vaccine-naïve participants and the number of BioNTech/Pfizer-vaccinated participants. Figure 1 and 2 show data for all participants as described in the legend, and Figure 3 for only one participant as defined in the figure legend. As this is not a genomic nor transcriptomic study, we cannot use the terminology of probands.

  1. I recommend summarizing the data on mucosal versus serum gene activities under different conditions and support this comparison by a graph or graphs that will help the reader to grasp the essential message of this article.

We agree that a summary figure will help the reader to follow the results of this proteomic study. We have added a figure in the new Conclusion section (Figure 6; lines 401-404) Due to the pilot nature of this research, we have focussed the summary figure on the different inflammatory profile of each compartment, the nasopharyngeal vs. the peripheral blood, and on the compartment-specific responses in vaccine-naïve and BioNTech/Pfizer-vaccinated participants.

  1. Similarly, a summary that clearly juxtaposes serum versus tissue activities will have to be added to the article. You cannot expect the reader to decipher each of the highly complex graphs for certain activities and arrive at  meaningful conclusions.

We have merged all the mucosal results in Figure 6 because we need further research to fully compare data from secretions and tissue.

Minor points:

Line 39 - Sentence should read: ..... could reveal correlates of protection ....

We have corrected the typo.

Figures 1c, 4a to c and 5 a to c: Please improve the graphic designation of genes. It is very difficult to decipher individual entries of nomenclature.

We apologize that during submission the quality of the images has been affected. We have added updated versions of the figures with higher quality of the protein names.

Reviewer 2 Report

The authors present a study shedding light on functional proteins of the mucosa tissue and secretions compared to systemically expressed ones. The study also explored the use of Olink technology for analyzing the presence of inflammation proteins in nasopharyngeal swabs. The study adds to the body of knowledge available on the topic. There are a few issues the authors would need to address with the manuscript particularly the presentation of results.

The authors write on lines 47-48 that no biomarkers of risk of protection have been defined in the mucosal portal of entry. I don’t think this is accurate since there are already a number of studies that have looked at this particularly with the protective aspect. The authors should consider rephrasing this.

On line 103, please define what BCA is.

Figure 1a. What is the unit on the NPx axis?. What are the parameters on the other axis? These should be clearly indicated.

Figure 1b. Units should be shown on scale. Axis labels should be provided.

Figure 1c. Axis labels are not legible and should be improved and provided where missing.

One of the goals of the study was to evaluate the use of Olink technology on nasopharyngeal swabs. Is there already an established method for doing this?. The values determined in this matrix by the Olink should be compared to values from another method for doing this for a proper comparison.

Figure 2a. Axis labels should be provided.

Figure 2b. Units should be shown on scale and axis labels provided.

Figures 4 and 5. Text on axis are not legible. Axis labels are also missing.

On line 74-76, other parameters including age and HIV-status was collected in this study but there is no mention of what this data was used for and does not show up in the correlation analysis. This should be addressed.

Is there any particular reason IgA was not included in the detected antibodies in the correlation analysis for mucosal samples at least?

Author Response

Response to Reviewer 2 Comments

The authors present a study shedding light on functional proteins of the mucosa tissue and secretions compared to systemically expressed ones. The study also explored the use of Olink technology for analyzing the presence of inflammation proteins in nasopharyngeal swabs. The study adds to the body of knowledge available on the topic. There are a few issues the authors would need to address with the manuscript particularly the presentation of results.

 The authors write on lines 47-48 that no biomarkers of risk or protection have been defined in the mucosal portal of entry. I don’t think this is accurate since there are already a number of studies that have looked at this particularly with the protective aspect. The authors should consider rephrasing this.

We thank the reviewer for this comment which will help improve the quality of the manuscript. We have amended the statement and added references (lines47-52).

On line 103, please define what BCA is.

We have defined the acronym of the assay (line 112).

Figure 1a. What is the unit on the NPx axis?. What are the parameters on the other axis? These should be clearly indicated.

The protein abundance measured with the Olink platform is expressed as NPx levels, these are the units of this technology. We have added a line in the methods to clarify this point (lines 130-131) and amended the x-axis title. The parameters shown on the y-axis are proteins, and this has been added in the figure.

Figure 1b. Units should be shown on scale. Axis labels should be provided.

Both have been added: y-axis is “Proteins” and x-axis are “Log2 fold change”.

Figure 1c. Axis labels are not legible and should be improved and provided where missing.

We apologize that during submission the quality of the image has been affected. We have added an updated version of the figure including axis titles.

One of the goals of the study was to evaluate the use of Olink technology on nasopharyngeal swabs. Is there already an established method for doing this?. The values determined in this matrix by the Olink should be compared to values from another method for doing this for a proper comparison.

Several methods can be applied to nasopharyngeal swabs for detection of proteins and a large number of studies in the field have been published applying mass-spectrometry. The present study highlights the possibility of measuring the proteomic profile of nasopharyngeal swabs using a technique, Olink, which is suitable for measuring proteomics in a large number of samples, in short time and at limited costs. A comparison between mass-spectrometry and O-link is not available as to work with over 90 protein targets in mass-spectrometry is not a realistic goal; in addition, the size of collected material would not allow for this comparison. We have added a paragraph in the discussion, however due to the high number of publications, we have added a few reviews on the subject (lines 341-348).

Figure 2a. Axis labels should be provided.

We have updated the figure to include the axis labels.

Figure 2b. Units should be shown on scale and axis labels provided.

We have updated the figure to include the axis labels.

Figures 4 and 5. Text on axis are not legible. Axis labels are also missing.

We apologize that the same problem mentioned above in Figure 1c has occurred with these figures. We have added an updated version of both figures including axis titles.

On line 74-76, other parameters including age and HIV-status was collected in this study but there is no mention of what this data was used for and does not show up in the correlation analysis. This should be addressed.

We agree that it would be interesting to evaluate the potential impact of all the baseline characteristics on the proteomic profile. In this brief communication we present initial analysis of the data collected. None of the participants enrolled were living with HIV and the age range was not broad enough to expect an impact the parameters measured. We intend to conduct further correlations analysis including more data. We have mentioned this in the discussion (lines 375-380).

Is there any particular reason IgA was not included in the detected antibodies in the correlation analysis for mucosal samples at least?

We agree that measuring the level of mucosal and/or systemic IgA would have been interesting.  However, we could only obtain qualitative assessment of systemic IgA titers. We have added this point in the paragraph of limitations in the Discussion (lines 395-397).

Round 2

Reviewer 1 Report

The authors have responded to the issues raised by this referee and have at least in major parts amended previously noted shortcomings of the manuscript.

They have also conceded that this is an intermittent report about an ongoing study. One can ask whether it would serve the authors and their readership if one waited for a more advanced stage of their study. However, considering that this manuscript contains already a great deal of detailed information, one could argue in favor of publishing at this stage. 

Would it be a fair compromise to limit this manuscript to the furthest advanced part of the study and leave some of the ongoing, less polished research work out and report it in a later, more focused communication. This referee would favor and recommend such a compromise. 

Author Response

Reviewer 1 comments

The authors have responded to the issues raised by this referee and have at least in major parts amended previously noted shortcomings of the manuscript.

They have also conceded that this is an intermittent report about an ongoing study. One can ask whether it would serve the authors and their readership if one waited for a more advanced stage of their study. However, considering that this manuscript contains already a great deal of detailed information, one could argue in favor of publishing at this stage.

Would it be a fair compromise to limit this manuscript to the furthest advanced part of the study and leave some of the ongoing, less polished research work out and report it in a later, more focused communication. This referee would favor and recommend such a compromise.

We appreciate the feedback of the reviewer. We agree that for a Research Article we would not include aspects such as the smoking status with is preliminary; however, considering that this is a Research Communication that intends to highlight to the reader the potential of the tissue explant models and multiplex technologies, such as Luminex and the Olink platform, for proteomic analysis of emerging viruses, we would be keen to keep the manuscript as it is. However, we have emphasized in the discussion the preliminary character of certain results (lines 319-320) and added a paragraph in the discussion emphasizing that we aim to show the potential of these technologies (lines 382-385). We hope it will be acceptable for the reviewer to maintain the manuscript as it is.

Reviewer 2 Report

I don't have any further comments.

Author Response

We thank the reviewer for approval.

Kind Regards

Round 3

Reviewer 1 Report

The authors have answered most of the concerns raised. One can still ask whether the amount of data will not overwhelm the potential readers. A more compact manuscript would probaby help. But of course, it remains the decision of the authors how they want to present their thoughts to the readers. Good luck.